# Social protection and informality in Latin America during the COVID-19 pandemic

**Matias Busso, Juanita Camacho, Julián Messina** [ID]**\*, Guadalupe Montenegro**

Research Department, Inter-American Development Bank, Washington, DC, United States of America

\* julianm@iadb.org

## Abstract

Latin American governments swiftly implemented income assistance programs to sustain families' livelihoods during COVID-19 stay-at-home orders. This paper analyzes the potential coverage and generosity of these measures and assesses the suitability of current safety nets to deal with unexpected negative income shocks in 10 Latin American countries. The expansion of pre-existing programs (most notably conditional cash transfers and non-contributory pensions) during the COVID-19 crisis was generally insufficient to compensate for the inability to work among the poorest segments of the population. When COVID-19 ad hoc programs are analyzed, the coverage and replacement rates of regular labor income among households in the first quintile of the country's labor income distribution increase substantially. Yet, these programs present substantial coverage challenges among families composed of fundamentally informal workers who are non-poor, but are at a high risk of poverty. These results highlight the limitations of the fragmented nature of social protection systems in the region.

**Data Availability Statement:** All data are publicly available household surveys.

**Funding:** The authors received no specific funding for this work.

## Introduction

At the beginning of the pandemic, governments around the world took immediate measures to contain the spread of the strain of coronavirus that has caused COVID-19, prioritizing in almost all cases some form of social isolation or distancing. The economic costs of these measures were not the same for everyone. The disease laid bare societies' inequalities, inflicting greater economic costs on the less economically fortunate [1]. Latin America, where working at home is a luxury that most people cannot afford [2–6], was no exception. Data from online surveys show that the impact of lockdowns across the region on job losses has been heavily concentrated in the bottom half of the income distribution [7]. These households to a large extent work in the informal sector, which excludes them from contributory government safety nets.

Adverse health shocks, such as a COVID-19 infection, are likely to affect the household's income and health both directly or indirectly. There is evidence that negative health shocks increase the risk of unemployment [8], can reduce the affected worker's earnings after recovery [8–10], and increase out-of-pocket expenses [11] thereby reducing household consumption of other goods and services. Negative income shocks, on the other hand, have been linked to

**Competing interests:** The authors have declared that no competing interests exist.

worse health outcomes both of children and adults [12–16]. The lockdown measures implemented by governments to stop the spread of a virus have been shown to cause hunger and worse nutrition [7], which are likely to increase inequalities in health outcomes [17], and exacerbate vulnerabilities to chronic diseases [18–20].

To overcome income losses among the most vulnerable households and workers, Latin American governments put in place a series of emergency social assistance programs. Governments additionally implemented a much larger set of emergency measures (from soft credits to formal firms to adjustments to monetary policy). This paper evaluates the potential impact of programs that target poor households and workers engaged in informal activities, which we broadly label as emergency social assistance programs. Among the 10 countries considered in this study, 33 ad hoc programs were launched between March and August 2020. For data availability reasons, the analysis focuses on Argentina, Bolivia, Brazil, Chile, Colombia, the Dominican Republic, Ecuador, El Salvador, Peru, and Uruguay. These countries cover 60 percent of the population in Latin America. Some of these programs were extensions of existing policies, most notably conditional and unconditional cash transfers and non-contributory pensions that expanded considerably during the first decade of the 2000s [21]. However, preexisting programs faced two limitations to reaching all households at risk during the lockdown. First, as is well known, their coverage is limited, even among the poor [22]. Second, the programs target the structurally poor and are not designed to mitigate temporary income shocks. Thus, many informal workers who are above the poverty line in regular times but would be severely affected by income losses during the lockdown are not eligible. To overcome these limitations, new ad hoc programs targeting specific groups (e.g., self-employed, and unemployed who are not eligible for unemployment benefits) were added to the set of emergency measures.

The emergency social measures studied are structured as (unconditional) monetary transfers. There is ample evidence that these programs can improve health outcomes through several channels. First, they increase food consumption and diet diversity among low-income beneficiaries [23]. Second, monetary transfers have also been shown to increase demand for preventive healthcare such as vaccinations [24] and clinic visits [25]. Third, there is evidence that transfer programs lead to better antropometric measures [25] and reduce the prevalence of illness (fever) among children [26]. Fourth, unconditional cash transfers have also been shown to improve mental heath. They have been associated, for instance, with reductions in depression among low-income young people [27, 28].

This paper provides an ex ante evaluation of the potential coverage and generosity of the COVID-19 emergency social assistance programs in 10 Latin American countries. The analysis maps the eligibility criteria of each program to the latest wave of nationally representative household surveys. We derive the coverage of the emergency programs and a measure of the potential replacement rate that compares the total transfer that each household may receive as a share of its regular labor income. The potential coverage and replacement rate are measured at different points of the income distribution to examine the distributional impacts of the programs.

We find that the potential coverage of the proposed emergency measures varies substantially by country, but in general it is fairly high among the poorest households that are in the first quintile of the earnings distribution, ranging from 54 percent in Ecuador to 100 percent (full potential coverage) in Brazil and Peru. Something similar happens with the replacement rate of potentially foregone labor incomes. With the exception of Uruguay, which introduced only selective lockdown in key sectors of economic activity and among vulnerable populations, the share of households with replacement rates below 25 percent in the first quintile of each country's earnings distribution does not exceed 20 percent. A different picture emerges in the

second and third quintiles, which in all cases except Brazil present much lower replacement rates.

The paper makes two main contributions to the literature. First, Latin America was a pioneer in the development of conditional cash transfer (CCT) programs. By now they are ubiquitous and have rapidly expanded to the rest of the world [29]. These programs provide income support to households and, at the same time, introduce incentives for attending school and demanding health services. Many studies analyze the effects of CCT programs on labor supply, human capital, and welfare (to name but a few [30–33] as well as features of their design (e.g., [34, 35]. Our paper assesses the suitability of these programs to respond to an unexpected crisis. It shows that the existing safety nets have the potential to be vertically expanded in times of crisis to make transfers more generous for those who are structurally poor and, thus, most likely to already be beneficiaries of those programs. At the same time, it demonstrates the limitations of those safety nets to be expanded horizontally to reach those who might fall into poverty temporarily. By doing so, the paper highlights the need for the region to develop a more robust system of automatic stabilizers that deal with the temporarily poor (e.g., unemployment insurance) that also accommodates the large existing informal economy [36, 37].

Second, this study contributes to the new but rapidly growing literature on government responses to the COVID-19 crisis. Much of this literature focuses on developed countries (e.g., [38]), although detailed analyses are available for Argentina [39] and Uruguay [40]. The analysis in this paper has implications for the feasibility of sustaining prolonged lockdowns of economic activity in Latin America. During the "new normality," while vaccination rates remain low, governments may have to impose capacity restrictions in many sectors to limit the spread of the virus (e.g., retail, restaurants). Moreover, self-enforcement and risk awareness have been key drivers of limited human interactions during the pandemic [41], suggesting that social distancing will continue until herd immunity is obtained, independently of government interventions. In this context, labor demand in occupations and sectors that require high physical proximity (like retail, hotels, restaurants, and many personal services) may be substantially dampened for a prolonged period of time. Because these occupations and sectors are intensive in low-skilled labor [1], they fundamentally employ informal workers in Latin America [5]. The challenges to sustaining the incomes of these workers and their families highlighted here will persist well into the recovery phase.

The rest of the paper is organized as follows. The first section outlines the problem of implementing effective social distancing measures in economies with high rates of informality, such as those in Latin America. The second section of the paper describes the emergency social assistance programs in place in each country and how they were mapped to identify potential beneficiaries in household surveys. The third section presents the core results by discussing the potential coverage and replacement rate of the emergency measures. The last section concludes by highlighting the main policy implications of our results.

## Social distancing in an informal economy

Most of Latin America implemented strict social distancing policies relatively early during the pandemic. By mid-March, the 10 countries under analysis in this paper had fewer than 100 COVID-19 cases [42]. Five days later, all 10 countries had closed their schools, and a week later there were strict lockdown measures in effect that required all non-essential businesses to close temporarily and their populations to stay home, in many cases under strict penalties [43]. The impacts of this widespread supply-side shock translated rapidly into a reduction of labor demand. Firm exits and job destruction amplified the initial effects of the lockdown, aggravating the recession [44]. In developing countries, the inability to telework, combined

with the high prevalence of labor informality imposed a natural limit to social distancing poli-
cies. [3] found that, for the sample of 10 countries used in this study, only 6.2 percent of indi-
viduals in the first quintile of the income distribution could work from home. The possibility
of teleworking was also limited for individuals in the second and third quintiles (8.4 and 10.9
percent, respectively). The limits on individuals teleworking are related to the task contents of
their jobs, sector of economic activity, size and sophistication of employers, and formality sta-
tus. Across occupations, only managers, professionals, technicians, and clerical workers are
more amenable to teleworking. Across industries, it is only in finance, insurance, real estate
and social services that the share of individuals who can telework exceeds 45 percent.

In contrast, only 6.7 percent of informal workers are estimated to be able to work from
home [3]. At the same time, informal workers have very little space to face unexpected income
losses. They have limited access to sick leave or unemployment benefits [45], they have on
average negative savings [46], and they have precarious access to health benefits. In the 10
countries under analysis, 41 percent of workers in the first and second quintiles of the labor
income distribution are self-employed. These workers most likely live hand-to-mouth. The
magnitude of the problem faced by policymakers in attempting to make social distancing feasi-
ble was made even larger because of assortative mating [47]. Table 1 shows the share of infor-
mal households. That is, those in which no household member contributes to social security,
independently of whether the member is an employee or self-employed. In all countries
included in the analysis, contributions to social security give workers the right to health insur-
ance. They also include contributions to a pension to be obtained after retirement, once other
eligibility criteria are met. On average, three in four households in the first quintile and more
than half in the second quintile of the labor income distribution are fully informal. These
shares decline for high-income households. This pattern is consistent across countries in the
region, but the share of informal households declines with the level of development. Among
the second quintile of each country's labor income distribution, more than 70 percent of the
households in Bolivia, Colombia, El Salvador, and Peru are fully informal. By contrast, the
shares are 24 and 18 percent in Chile and Uruguay, respectively. Informality imposes an addi-
tional social cost when households face generalized and unexpected income losses. In a context

**Table 1. Percentage of households without formal workers, by country and income quintile.**

|  | Q1 | Q2 | Q3 | Q4 | Q5 |
|---|---|---|---|---|---|
| Argentina | 74 | 44 | 29 | 23 | 19 |
| Bolivia | 97 | 86 | 75 | 61 | 47 |
| Brazil | 67 | 31 | 19 | 14 | 9 |
| Chile | 46 | 24 | 16 | 14 | 11 |
| Colombia | 94 | 70 | 44 | 34 | 17 |
| Dominican Republic | 69 | 51 | 43 | 41 | 32 |
| Ecuador | 83 | 63 | 48 | 34 | 20 |
| El Salvador | 94 | 72 | 54 | 45 | 34 |
| Peru | 99 | 87 | 68 | 51 | 38 |
| Uruguay | 51 | 18 | 8 | 4 | 2 |
| LAC | 77 | 55 | 40 | 32 | 23 |

Note: Unweighted average for LAC. Data are from 2018 household surveys from Inter-American Development
Bank–Harmonized Surveys for LAC, except Chile (2017). Income quintiles are calculated at the household level using
monetary labor income per capita. LAC = Latin America and the Caribbean.

of sudden crisis, full household informality severely limits the possibility of consumption smoothing through within-household risk-sharing mechanisms [48–50].

Recognizing that social distancing would not be feasible without some sort of income support program, countries in Latin America moved swiftly to compensate households for their potential lost income.

## Approximating government emergency social assistance programs

Our analysis is based on two main sources of data. First, we collected and coded COVID-19 emergency assistance measures that were identified primarily through official government websites that track policy responses. The information in these websites has sometimes been incomplete or updated with some delay. To complete and validate this information, we used three sources. First, the main newspapers in each country were scraped, searching for specific keywords. We searched newspaper websites for the words "subsidy," "transfer," "coronavirus," and "aid" (in Spanish or Portuguese). Second, we checked the "Weekly policymakers response against COVID-19 database" compiled by the COVID-19 Policy Measures Team at the Inter-American Development Bank. Third, we checked the "ACAPS COVID-19: Government Measures Dataset." All sources are included in the appendix in S1 File. Our second sources of data are household surveys from 2018 that were used to match the descriptions of the programs to household or individual characteristics in the survey. Except in Chile, where the National Socio-Economic Characterization Survey (CASEN) is bi-annual and the latest available wave was 2017. See the appendix in S1 File for a full list of the surveys used.

Table 2 shows a detailed view of the policies implemented by the 10 countries. The transfers are identified by the name the country gave them (e.g., Bono Universal in Bolivia) or by the beneficiaries targeted for the cash transfer. We identified the details of each policy (as described in the law or government announcement), targeted beneficiaries (households or individuals), amount and frequency of cash transfers, and eligibility criteria. A total of 33 programs were put in place in the 10 countries. More than half of these programs (18) were extension of existing policies, most notably conditional cash transfers and non-contributory pensions.

There is limited information on the budget allocation to each of these programs and their sources of financing, but all countries increased significantly their budgets for social protection during the early stages of the pandemic. The raise of budget on social protection between 2019 and 2020 ranges from 8 percent in Uruguay to 225 percent in Dominican Republic (Table 3).

Table 4 shows the criteria used to identify the "full lockdown" periods in every country. The start date is the date on which the country was under a national lockdown mandate, with three exceptions. Chile and Brazil never implemented a national lockdown, instead following specific indicators to undertake selective lockdowns of specific regions (Chile) or leaving the final decision to each state (Brazil). In these two cases we consider the start date of the lockdown when the largest state/region of each country, Santiago and Sao Paulo, implemented a full lockdown. Because the programs discussed in the paper potentially are national (or federal) programs, in Brazil and Chile we have produced two sets of simulations, one for the entire country and the other for the regions of Sao Paulo and Santiago, respectively. The other exception is Uruguay, which contained the spread of the virus early on and never imposed a full national lockdown. In Uruguay we chose as start date the time at which all non-essential business were shut down. The end date was less clear-cut because each country implemented different progressive reopening plans. We chose as the end date the moment when sectors such as construction and retail were allowed to resume operations. We only take into account

**Table 2. Emergency social assistance measures.**

| Country | Policy | Beneficiary description | Transfer Level | Pre-existent Social program |
|---|---|---|---|---|
| **Argentina** | 1 | Retirees, pensioners, and noncontributory pension beneficiaries receiving up to ARS$18,892 for their monthly pension | Individual | Yes |
| | 2 | Beneficiaries of Universal Child Allowance (AUH) | Household | Yes |
| | 3 | Per pregnant female at the household (AUE) | Individual | Yes |
| | 4 | Tarjeta Alimentar—for parents of children affiliated to AUH who are not over age 6 years | Household | Yes |
| | 5 | Tarjeta Alimentar—for pregnant women in their third trimester or more who have the AUE benefit | Household | Yes |
| | 6 | Ingreso familiar de emergencia—transfer for households with a household head between 18 and 65 who works in domestic service, is an informal worker, is a *monostributista social* (categories A and B), or households receiving AUH or Progresar social programs; household must not have a formal source of labor income or receive any pensions | Household | No |
| **Bolivia** | 7 | Bono Familia—transfer per child enrolled in school (does not include tertiary level) | Individual | No |
| | 8 | Bono Universal—for adults between ages 18 and 60 who do not receive any other government transfers (for retirement, widowhood, disability or meritorious), nor wages from public or private institutions, nor pensions or rents | Individual | No |
| | 9 | Canasta Familiar—transfer for older adults who receive Renta Dignidad but no other rent or pension; mothers who receive the Juana Azurduy transfer; or people with disability who receive the disability bonus | Individual | Yes |
| **Brazil** | 10 | Transfer for households with a single mother as household head, or with individuals whose main source of income comes from being informal workers or self-employed; unemployed; or microentrepeneurs; these households must not be beneficiaries of Bolsa Familia; their total income must not be more than R$3,135 or total per capita income above R$522.5 | Household | No |
| | 11 | Beneficiaries of Bolsa Familia who do not receive other benefits; their total income must not be more than R$3,135 or total per capita income above R$522.5 | Household | Yes |
| **Colombia** | 12 | Beneficiaries of Familias en Accion | Individual | Yes |
| | 13 | Beneficiaries of Jovenes en accion | Individual | Yes |
| | 14 | Beneficiaries of Colombia Mayor | Individual | Yes |
| | 15 | Ingreso Solidario—Households under extreme poverty, poverty, or economic vulnerability that do not receive any social program (Familias en accion, Jovenes en accion, Colombia Mayor) but belong to SISBEN | Household | No |
| **Chile** | 16 | Ingreso Familiar de Emergencia—transfer for households whose source of income is mainly from informal sources. The amount depends on the number of people in the household and decreases according to the percentage of income that is formal; pensioners from Pension Solidaria de la Vejez receive a smaller amount of aid | Household | No |
| | 17 | Bono Invierno—transfer for older adults who do not receive more than one pension or whose amount received is less than CLP$166,191 and who are retired from specific institutions (Instituto de Prevension Social, Instituto de Seguridad Laboral, Direccion de Prevision de Carabineros de Chile, Caja de Prevision de la Defensa Nacional, among others) or if they are beneficiaries of the program Pension Basica Solidaria de Vejez | Individual | Yes |
| | 18 | Bono de Emergencia COVID 19—this transfer aims at households with individuals receiving Subsidio Familiar (SUF), households in the Sistema de Seguridades y Oportunidades database, households who belong to the 60% most vulnerable according to the Registro Social de Hogares database, and households who do not have a formal income through employment or pension and do not have any SUF beneficiaries | Household | Yes |
| **El Salvador** | 19 | Transfer for informal employees and self-employed workers with low social economic resources | Household | No |
| **Ecuador** | 20 | Transfer for affiliates to the unpaid work regime or self-employed; or affiliates to the Seguro Social Campesino, with income less than US$400 and who are not registered to the contributive social security and are not registered as dependents; individuals must not be beneficiaries of any other programs of the government | Individual | Yes |
| | 21 | Transfer for people not included in the previous subgroup whose income is lower than $400 and are below the poverty line | individual | No |

(*Continued*)

**Table 2.** (Continued)

| Country | Policy | Beneficiary description | Transfer Level | Pre-existent Social program |
|---------|--------|-------------------------|----------------|------------------------------|
| **Peru** | 22 | Bono Quedate en Casa—Transfer for urban households below poverty line, who are not beneficiaries of Pension 65 or Juntos | Household | No |
| | 23 | Bono Independiente—transfer for households with main income source coming from self-employment and not in poverty; households cannot be beneficiaries of the Juntos, Pension 65, or Contigo programs; none of the household members can be registered as dependent workers of the public or private sector; household members cannot have income over PEN$1,200 and cannot be part of any local or central government | Household | No |
| | 24 | Bono Rural—transfer for rural households below poverty line, who are not beneficiaries of Pension 65 or Juntos | Households | No |
| | 25 | Bono Familiar Universal—transfer for households in poverty and ex-treme poverty; beneficiaries of the Juntos, Pension 65, or Contigo programs, and households above the poverty line and having no members registered as dependent workers of the public or private sector; none of the household members can have income greater than PEN $3,000; and only households who have not received previous transfers from COVID-19 aid (Policies (22) to (24) can receive this transfer | Household | Yes |
| **Dominican Republic** | 26 | Beneficiaries of the Solidaridad social Comer es Primero program | Household | Yes |
| | 27 | Transfer for households who do not have any Solidaridad program Comer es Primero benefits and are under poverty and vulnerability according to SIUBEN | Household | No |
| | 28 | Additional transfer for groups in Policies (28) and (29) whose household head is vulnerable (age 60 + years) | Household | Yes |
| **Uruguay** | 29 | Extra transfer for Tarjeta Uruguay Social beneficiaries | Household | Yes |
| | 30 | Transfer for adults who are 65+ years and still working in the private sector (sickness benefit due to quarantine measures) | Individual | No |
| | 31 | Beneficiaries of Plan Equidad | Household | Yes |
| | 32 | Transfer for food purchases for informal and self-employed workers, with no other social program benefits and who do not have social security | Individual | No |
| | 33 | Transfer for a certain type of taxpayers (monotributistas sociales del MIDES) | Indiviidual | No |

**Table 3. Government expenses on social protection by country in 2019 and 2020 (LCU in millions).**

| | 2019 | 2020 | Var % |
|---|------|------|-------|
| Argentina | 2,218,102 | 3,614,937 | 63.0 |
| Brazil | 941,702 | 1,301,381 | 38.2 |
| Chile | 11,999,419 | 16,286,275 | 35.7 |
| Colombia | 10,243,156 | 17,476,454 | 70.6 |
| Dominican Republic | 57,493 | 187,117 | 225.5 |
| Ecuador | 1137 | 1558 | 37 |
| El Salvador | 2,569 | 2,876 | 12 |
| Peru | 6,613 | 14,046 | 112.4 |
| Uruguay | 19,788 | 21,445 | 8.4 |

Note: The numbers refer to total expenses on "Social Inclusion and Reconciliation" (Colombia), "Social Security" (Argentina), "Social Development" (El Salvador), "Social Benefits" (Dominican Republic and Ecuador) and "Social Protection" (Peru, Chile, Brazil and Uruguay). Sources: Transparency Portal (Colombia, Peru and Uruguay), Fiscal Observatory foundation (Chile), reports by the Ministry of Finance (Dominican Republic and Argentina), National Treasury (Brazil), and Ministry of Economy and Finance (Ecuador, Uruguay and El Salvador). We do not include Bolivia in the table as we did not find updated information on social protection expenses.

**Table 4. Dates of full lockdown.**

| Country | Start date | Criteria | End date | Criteria |
|---|---|---|---|---|
| Argentina | 3/20/2020 | Complete national lockdown. Only essential business is open. | 6/8/2020 | Non-essential businesses in Buenos Aires reopen, while reopening plans are underway in other parts of the country under different social distancing measures. |
| Brazil | 3/24/2020 | Sao Paulo goes on lockdown. Only essential business is open. | 6/10/2020 | Non-essential business start reopening in Sao Paulo under social distancing rules. |
| Bolivia | 3/20/2020 | Complete national lockdown. Only essential business is open. | 6/1/2020 | Zonal reopening phase starts. In five departments most non-essentials business are permitted. Four departments continue on lockdown. |
| Chile | 3/25/2020 | In Santiago, certain sectors of the economy and schools closed down by 3/21. National curfew from 10 pm until 5 am starts on 3/22. Zonal lockdown in several neighborhoods start on 3/26. | 8/17/2020 | First neighborhoods of Santiago start reopening. |
| Colombia | 3/25/2020 | Complete national lockdown. Only essential business is open. | 6/1/2020 | "Intelligent isolation" begins. Malls and shops reopen, but only for commercial (no social) activities. |
| Ecuador | 3/17/2020 | Complete national lockdown comes into place. Only essential business is open. | 5/4/2020 | Progressive reopening under social distancing. |
| El Salvador | 3/21/2020 | Complete national lockdown. Only essential business is open. | 6/16/2020 | Phase 1 of the reopening plan starts. Some non-essential sectors are allowed to go back to business. |
| Peru | 3/15/2020 | Complete national lockdown. Only essential business is open. | 6/5/2020 | Phase 2 begins. Some non-essential sectors are allowed to go back to business. |
| Dominican Republic | 3/17/2020 | Travellers must quarantine, several social distancing measures and non-essential sector closures come into place. Schools close down. | 6/3/2020 | Phase 2 begins. Most non-essential business is allowed to resume activities. |
| Uruguay | 3/13/2020 | Social distancing is recommended (people are adviced to stay home). Schools and non-essential businesses close. | 4/13/2020 | Social distancing phase begins. Construction sector resumes activities and some rural schools reopen. |

emergency programs that start during the full lockdown period in each country, as well as disbursements that were programmed during this time.

We then relied on national household surveys to match the eligibility criteria of each program with pre-existing household characteristics. In 19 programs the identification of potential beneficiaries in the household surveys was straightforward. This was the case for programs that expanded existing policies whose beneficiaries were already identifiable in the surveys, such as Familias en Accion in Colombia. Another 12 policies were reasonably approximated with survey respondents' characteristics. Two programs in Argentina target pregnant women, a characteristic we cannot observe in the surveys. Hence, these two programs have been excluded from the analysis. The details of the mapping between programs and household survey characteristics are provided in the Appendix. Once potential beneficiaries were identified, we calculated potential coverage and replacement rates by household labor income quintile in each country. A household is defined as covered if at least one member was targeted by at least one program during the lockdown period. The replacement rate is defined as a ratio between all the aid a household may potentially receive per month and the household's typical monthly labor income (as measured in 2018, but updated to 2020 prices using national consumer price indices). Households whose labor income was zero or negative in the survey were excluded from the analysis. When the program provided for a lump-sum stipend for the duration of the lockdown we transformed the stipend into the equivalent monthly pay. This replacement rate is indicative of the ability of the emergency measures to replace households' foregone labor incomes assuming that, during the lockdown period, those incomes were negligible. It is plausible, of course, that some workers found, during the period of full lockdown, alternative sources of income. Unfortunately, that information is not available. Our estimates should be

interpreted as an upper bound of the actual coverage and replacement rates in each country for two reasons. First, we focus on potential rather than actual beneficiaries. Implementing these programs with complex eligibility criteria during a pandemic was challenging in most countries. Similarly challenging was the ability to receive aid in a region where less than 40 percent of the population has a bank account and access to financial services is particularly low among households in the bottom half of the distribution [51]. Despite these limitations, some governments attempted to expand the reach of the programs. For example, beneficiaries of several emergency programs in Colombia could opt to be enrolled in mobile wallet platforms and bank accounts to receive the aid. Second, while in a few cases the program eligibility excluded certain parts of the population (usually because they were beneficiaries of other transfer programs), we nonetheless had no information in the household survey to identify certain eligibility criteria of a program. In those cases we assumed that this exclusion was not imposed. For example, the "Bono Independiente" in Peru targeted households above the poverty line, where members were not employed formally and were not enrolled in either the "Juntos" or the "Pension 65" social programs. All these traits are observable in the data. However, targeted households could not be beneficiaries of the "Contigo" program, which we do not observe. This restriction was ignored.

## Coverage and generosity of the transfers

Fig 1 summarizes the main results of the paper. It shows two indicators of the potential coverage of emergency transfers, measured as the share of households that are expected to receive them. One measure includes only emergency transfers that are extensions of programs that existed before the COVID-19 crisis, and the other considers all emergency transfers. The third line shows the generosity of the emergency programs combined, calculated as the median emergency transfer received by households each month as a proportion of their regular monthly household labor income. The three measures are computed by quintile of the household labor income distribution of each country and averaged across the 10 countries.

The emergency transfers had the potential to reach a high proportion of households in the first quintile, but coverage dropped linearly with earnings. This left a substantial share of households in the second and third quintiles uncovered, although these quintiles have a high concentration of fully informal households (see Table 1). The high coverage in the first quintile, potentially reaching 85 percent of households, hinged on the set of ad hoc measures introduced by governments in the region. If governments were to rely only on the expansion of existing social programs, the coverage would have been much lower (about 45 percent). This excludes El Salvador, which opted not to rely on its CCT program to cover the earning losses of the COVID-19 lockdown measures. The extension of the ad hoc programs also shows potential leakages. Up to 25 percent of the households in the richest quintile could become beneficiaries of one of these emergency social assistance programs.

Table 5 shows the share of households in each quintile of the earnings distribution of each country that was expected to receive an emergency social assistance transfer. Panel A considers the emergency transfers that were allocated to households via preexisting social programs. On average, 45 percent of the households in the first earnings quintile received an emergency transfer through the preexisting infrastructure of the social safety net, but this varies substantially across countries, with coverages as low as 5 percent in Ecuador and as much as 80 percent in Brazil. This is not surprising, given the low coverage of safety nets in the region. [22] report that on average CCT programs in Latin America cover about 43 percent of households below the poverty line who have children. Similarly, noncontributory pension programs cover about 46 percent of the elderly who are under the poverty line. The reasons for low coverage

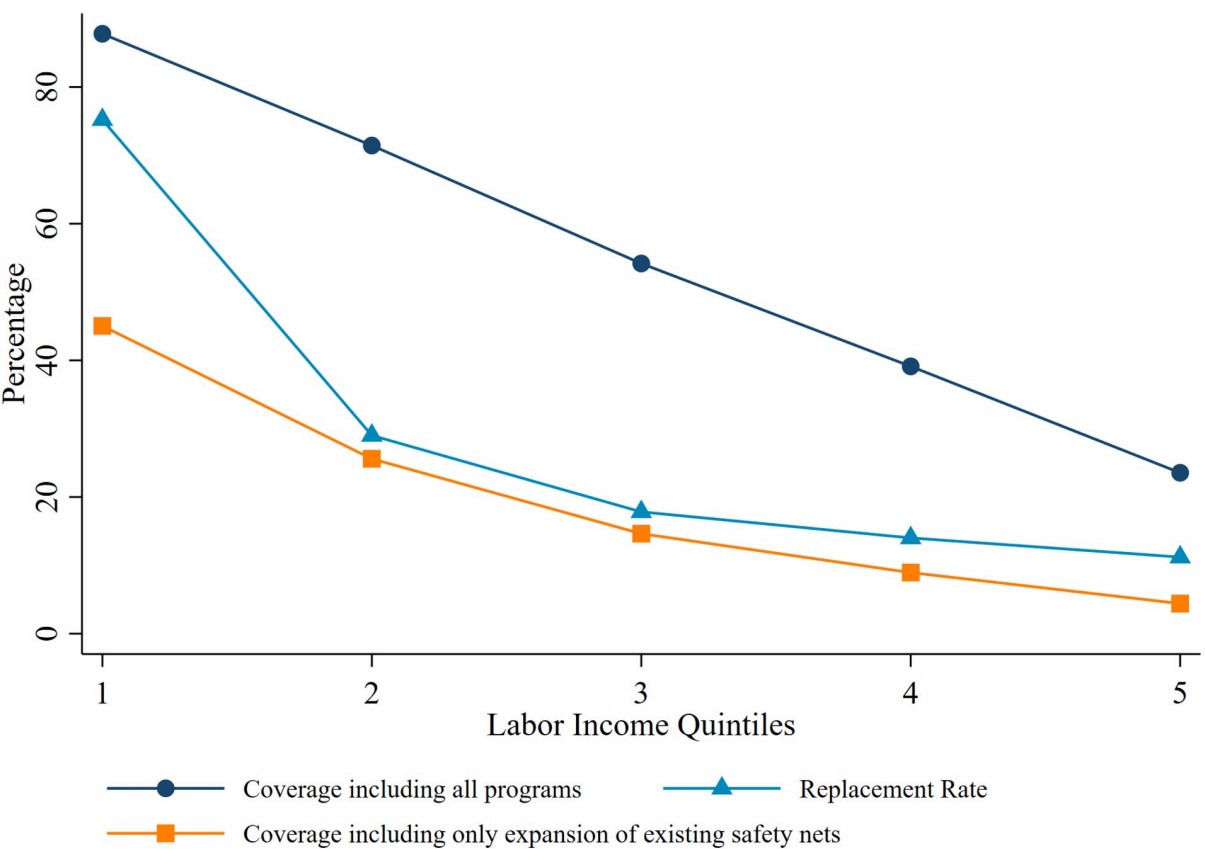

**Fig 1. COVID-19 emergency social assistance programs in LAC: Coverage and replacement rates.** Note: Unweighted average for LAC. (i) Coverage is defined as the percentage of household receiving aid (ii) Replacement Rate is the median of the monthly monetary transfer over the monthly monetary labor income for the targeted households. LAC average for coverage including only expansion of existing safety nets does not include El Salvador.

are potentially many. Some of the programs are small due to fiscal constraints; targeting is based on proxy-means testing, which is imperfect; lack of connection or distrust of social services; and inability to comply with the eligibility criteria or conditionalities. Interestingly, [22] report that, among the CCT and non-contributory pension programs in the region, the country with the lowest coverage is El Salvador, reaching as little as 11 percent of poor households with children and 9.4 of the elderly who are poor. El Salvador has been the only country among those considered in this study that did not use targeting of its preexisting safety net to provide emergency transfers during the COVID-19 pandemic.

Being aware of the coverage limitations of preexisting social assistance programs, governments around the region expanded the eligibility criteria with a set of ad hoc measures. Table 5, panel B, shows the coverage of all emergency transfers, which reached more than 90 percent of households in the first quintile in six of the 10 countries. Coverage was lower in most countries for households in the second or third quintiles. In Bolivia, Brazil, El Salvador, Peru, and Uruguay, however, it reached on average 84 percent of the second and third quintiles, which allowed the emergency transfers to provide assistance to the lower-middle class. Simulations discussed in the text refer to Brazil and Chile as a whole. The results for Sao Paulo and Santiago suggest lower coverage (results were calculated at the regional level, Sao Paulo State for Brazil and Region Metropolitana de Santiago for Chile). In Sao Paulo, the coverage of

**Table 5. Percentage of targeted households by type of monetary transfer, country and income quintiles.**

| | A. Preexisting social programs | | | | | B. All transfers | | | | |
|---|---|---|---|---|---|---|---|---|---|---|
| | Q1 | Q2 | Q3 | Q4 | Q5 | Q1 | Q2 | Q3 | Q4 | Q5 |
| Argentina | 65 | 46 | 34 | 20 | 7 | 70 | 52 | 41 | 29 | 15 |
| Bolivia | 50 | 28 | 24 | 22 | 16 | 93 | 94 | 90 | 81 | 50 |
| Brazil | 80 | 23 | 0 | 0 | 0 | 100 | 97 | 68 | 21 | 4 |
| Chile | 32 | 20 | 12 | 7 | 4 | 97 | 58 | 41 | 27 | 20 |
| Colombia | 38 | 23 | 12 | 5 | 1 | 88 | 55 | 12 | 5 | 1 |
| Dominican Republic | 39 | 30 | 23 | 17 | 9 | 84 | 49 | 23 | 17 | 9 |
| Ecuador | 5 | 6 | 3 | 2 | 1 | 54 | 44 | 41 | 35 | 17 |
| El Salvador | 0 | 0 | 0 | 0 | 0 | 96 | 83 | 80 | 76 | 64 |
| Peru | 46 | 23 | 9 | 3 | 1 | 100 | 93 | 71 | 50 | 26 |
| Uruguay | 51 | 31 | 13 | 4 | 1 | 97 | 90 | 74 | 51 | 29 |
| LAC | 45 | 26 | 15 | 9 | 4 | 88 | 71 | 54 | 39 | 24 |

Note: LAC = Latin America and the Caribbean. Unweighted average for LAC. 2018 household surveys from IDB-Harmonized Surveys for LAC, except Chile (2017). Income quintiles are calculated at the household level using the distribution of monetary labor income per capita in each country. Panel A shows the percentage of targeted households receiving monetary transfers if countries only used preexistent social programs. The LAC average in panel A does not include El Salvador. Panel B shows the percentage of targeted households including all the policies implemented. The LAC average in panel B includes all 10 countries.

pre-existing social safety nets potentially reached 56% (0%) of the households in the first(second) quintile. When all programs are considered together, the coverage increases to 100% and 79% in the first and second quintiles, respectively. In Chile, the coverage of existing safety nets in the first(second) quintile was 27%(16%), jumping to 88%(50%) when considering all policies.

Table 6 assesses the ability of the emergency transfers to replace potential labor income losses. Panel 6A shows the median replacement rate, and panel 6B shows the share of

**Table 6. Replacement rate of COVID-19 emergency social assistance by country and income quintile.**

| | A. Median | | | | | B. Less than 25% | | | | |
|---|---|---|---|---|---|---|---|---|---|---|
| | Q1 | Q2 | Q3 | Q4 | Q5 | Q1 | Q2 | Q3 | Q4 | Q5 |
| Argentina | 56 | 22 | 14 | 9 | 6 | 16 | 63 | 94 | 100 | 100 |
| Bolivia | 84 | 19 | 10 | 6 | 4 | 7 | 69 | 97 | 100 | 100 |
| Brazil | 164 | 79 | 57 | 55 | 57 | 0 | 0 | 0 | 0 | 0 |
| Chile | 41 | 17 | 9 | 8 | 2 | 32 | 67 | 89 | 98 | 100 |
| Colombia | 38 | 16 | 9 | 5 | 3 | 24 | 85 | 99 | 99 | 100 |
| Dominican Republic | 49 | 26 | 16 | 13 | 9 | 9 | 48 | 72 | 82 | 95 |
| Ecuador | 99 | 35 | 15 | 10 | 6 | 3 | 39 | 74 | 85 | 100 |
| El Salvador | 67 | 36 | 24 | 18 | 11 | 2 | 22 | 55 | 74 | 89 |
| Peru | 129 | 30 | 18 | 13 | 12 | 1 | 35 | 71 | 85 | 88 |
| Uruguay | 18 | 6 | 3 | 2 | 1 | 63 | 94 | 94 | 94 | 87 |
| LAC | 74 | 29 | 18 | 14 | 11 | 16 | 52 | 74 | 82 | 86 |

Note: LAC = Latin America and the Caribbean. Unweighted average for LAC. Data are from 2018 household surveys from IDB-Harmonized Surveys for LAC, except Chile (2017). Income quintiles are calculated at the household level using monetary labor income per capita. The replacement rate is defined as total monthly transfer divided by regular monthly labor income in the household. Non-targeted households by the emergency programs and households with zero or negative regular labor income are excluded. Panel A shows the median of the replacement rate over the monetary labor income for targeted households. Panel B shows the percentage of targeted households for which the replacement rate is less than 25%.

households for which the emergency transfer would replace less than 25 percent of their pre-COVID-19 earnings. Households not targeted by emergency programs and households with zero or negative regular labor income are excluded from the calculations.

The replacement rate was very high in the first quintile of all countries, in some cases more than compensating for the potential labor income loss. A notable exception is Uruguay (potential replacement rate of 18 percent). Similarly, except in Uruguay, Chile and Colombia, the share of households in the first quintile of the earnings distribution with a replacement rate below 25 percent was lower than 20 percent.

The transfers replaced only a small share of the potential earning losses of households in the second and third quintiles. Brazil is the only exception. In the other nine countries, the median replacement rate did not reach 50 percent among households in the second quintile, and it was below 30 percent in the third quintile. The emergency programs also left a substantial share of households with replacement rates below 25 percent. More than 50 percent of households were below this threshold in the second quintile in five of the 10 countries, and more than 80 percent in the third quintile would receive a transfer that replaced less than 25 percent of their labor income if it were to become zero during the lockdown. Having adopted social distancing measures later than most of the countries in the group, Brazil is an outlier. Potential beneficiaries in Brazil in all quintiles of the distribution greatly benefited from the transfers. The two programs included in the emergency social assistance measures have a median replacement rate that was up to 57 percent of regular labor income in the richest quintile, leaving no household with a potential transfer that would cover less than 25 percent of its regular labor income. Simulations for Sao Paulo and Santiago also suggest lower replacement rates in the first and second quintiles. The median replacement rate for Sao Paulo was 98% and 59% in the first and second quintiles, respectively, while for Santiago it was 37% and 11%. The difference between the national and the regions' replacement rates is smaller when considering the fourth and fifth quintiles: 57% and 46% in Sao Paulo, and 6% and 2% in Santiago, respectively.

## Conclusion

At the onset of the COVID-19 pandemic, Latin American governments took aggressive steps to save lives by imposing shelter-in-place measures to stop the propagation of the virus. In most cases, they swiftly implemented compensation programs to sustain incomes and facilitate the stay-at-home orders. This paper highlights the strengths and limitations of these emergency programs by analyzing their potential coverage and generosity in 10 Latin American countries. In doing so, the paper assesses the suitability of current safety nets in the region to deal with unexpected systemic negative income shocks.

The proposed emergency measures were able to target the most vulnerable households: those in the first quintile of the country's labor income distribution. However, the coverage and replacement rates in the second and third quintiles were much more limited. With the notable exceptions of Brazil and Bolivia, emergency social assistance programs as currently formulated could not replace the potentially foregone incomes of a large fraction of families in the informal workforce that were forced to shelter in place and could not work. This insufficient compensation has surely limited the ability of governments to sustain extended lockdown periods, and it may limit the ability to enforce another wave of lockdowns if there were a resurgence of contagion.

Government responses to the COVID-19 crisis highlight a major structural problem in the region: the fragmented and insufficient coverage of social protection systems. These limitations complicate the delivery of income support to informal workers who are not sufficiently poor to benefit from social assistance but lack other automatic stabilizers, such as

unemployment insurance, that could alleviate the impact of temporary shocks. Although the answer to how to protect those who do not contribute to social protection systems is not straightforward, the need to have mechanisms that reach these households during times of negative shocks has become evident. Countries should generalize their information systems to identify the near poor, or vulnerable populations, updating the information on a regular basis. The feasibility and budgetary needs of programs that activate guaranteed minimum income schemes during recessions merits more study in the region. Programs that tackle idiosyncratic income risk will be much more difficult to implement while the levels of informality in the region remain high.

The full impact of the social protection systems utilized during the course of the current outbreak is still unknown. Latin America needs to consider reforms that would provide more effective and agile assistance to those falling through the cracks in times of crisis. Doing so would make the region more resilient in the wake of future negative shocks.

## Supporting information

**S1 File. Appendix: Identifying COVID-19 emergency measures and lockdown periods.** (ZIP)

**S2 File. Replications files.** Includes do-file and STATA dta to replicate results. (7Z)

## Acknowledgments

We thank Eric Parrado and Norbert Schady for comments on an early draft as well as participants in several seminar presentations. All errors are our own. The opinions expressed in this publication are those of the authors and do not necessarily reflect the views of the Inter-American Development Bank, its Board of Directors, or the countries they represent.

## Author Contributions

**Conceptualization:** Julián Messina, Guadalupe Montenegro.

**Data curation:** Juanita Camacho, Guadalupe Montenegro.

**Formal analysis:** Juanita Camacho, Guadalupe Montenegro.

**Writing – original draft:** Matias Busso, Julián Messina.

**Writing – review & editing:** Matias Busso, Julián Messina.

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
