## [Decision Letter · Decision Letter 0]

30 Jun 2021

PONE-D-21-12981

Social Protection and Informality in Latin America during the COVID-19 Pandemic

PLOS ONE

Dear Dr Messina and colleagues,

Thank you for the opportunity to review this interesting, important, well executed, and insightful piece on cash transfers in latin america during Covid-19 times. I am recommending the piece for publication following minor revisions.

Both reviewers reported the piece to be strong and a good addition to the literature. You will note that reviewer 1 felt that the piece may be better suited to another journal. Following clarification with the PLOS One editorial office, we feel that the piece meets the publication criteria of PLOS One but I would like you to add (perhaps as a paragraph/section) a brief critical appraisal of the evidence relating to the health and public health impacts of the cash transfer schemes, including on health inequalities if possible. We think that this would strengthen the manuscript and make it even more relevant for PLOS One readership. It would also be beneficial to researchers and policy makers in the field of medicine, especially poverty-related diseases such as tuberculosis, to have a summary of this evidence with which to guide future strategies.

Please do not hesitate to contact us if you require further information and many thanks for submitting to PLOS One.

We look forward to receiving your revised manuscript.

Kind regards,

Tom E. Wingfield

Academic Editor

PLOS ONE

Journal Requirements:

4. Thank you for submitting the above manuscript to PLOS ONE. During our internal evaluation of the manuscript, we found significant text overlap between your submission and the following works:

- https://cepr.org/sites/default/files/CovidEconomics27.pdf

Please revise the manuscript to rephrase the duplicated text, cite your sources, and provide details as to how the current manuscript advances on previous work. Please note that further consideration is dependent on the submission of a manuscript that addresses these concerns about the overlap in text with published work.

Reviewers' comments:

Reviewer's Responses to Questions

**Comments to the Author**

1. Is the manuscript technically sound, and do the data support the conclusions?

Reviewer #2: Yes

Reviewer #3: Yes

2. Has the statistical analysis been performed appropriately and rigorously? 

Reviewer #2: Yes

Reviewer #3: Yes

3. Have the authors made all data underlying the findings in their manuscript fully available?

Reviewer #2: Yes

Reviewer #3: Yes

4. Is the manuscript presented in an intelligible fashion and written in standard English?

Reviewer #2: Yes

Reviewer #3: Yes

5. Review Comments to the Author

Reviewer #2: Dear authors,

thank you for submitting this very useful paper. The paper has the potential to inform a number of additional studies on the extent to which social protection has indeed been a part of the pandemic response and has contributed to its control by enabling people to respect and be more compliant with the imposed social distancing measures.

Nonetheless, I think this paper seems better suited in either an economy or development Journal rather than Plos.

I think that in order to meet better the requirements of this journal there should be a more explicit health focus or at least a focus on the public health impact of the interventions you discuss.

I wonder whether your analysis could be complemented to meet either of the above, for example by attempting to see whether the coverage and income replacement achieved correlate with the severity of the pandemic (in terms of morbidity and mortality) or even the duration of it. In other words, in my opinion what is missing in the paper is a more obvious link with either the containment of the pandemic or the mitigation of the health damaging effect of the control measures. For example, authors could try to link the data they present (even only ecologically) with the exacerbation of health inequalities or some potential excess in the burden of other diseases (resulting from the social distancing measures) in these countries. A research question could be: do country with a better income replacement rate show also better covid morbidity and mortality or more stable health indicators (i.e. suicide, alcohol use, violence and other indicators that are sensitive to financial shocks).

In conclusion, this is an extremely interesting and potentially useful paper. I just don't think Plos is a good fit for it or it could become a good fit by adding a third piece of analysis able to look at how the findings of this paper (in terms of coverage and income replacement) correlate with a better or worse success in the management of the pandemic.

The above could be done for all countries ideally, or pick up one or two countries to be more in depth case studies.

Hope this will be helpful - I remain available to discuss further how the paper could be more relevant in the public health sphere.

Best

Reviewer #3: This paper is well written and methodologically sound. It makes an important contribution to the literature on social protection coverage and shock-responsive social protection. The multi-country approach is a strong aspect of this paper.

Introduction

1. The following sentence should be changed to reflect that informal sector work excludes populations from “contributory” government safety nets. Indeed, many large social safety nets in LAC which are non-contributory (Bolsa Familia, PROSPERA, etc. do include informal sector workers. “These households to a large extent work in the informal sector, which excludes them from government safety nets.”

2. Authors could add language on vertical expansion (to existing participants) and horizontal expansion (to those who might fall poor) in their discussion of potential coverage on page 2: “potential to be expanded in times of crisis to make transfers more generous for those who are structurally poor. At the same time, it demonstrates the limitations of those safety nets to reach those who might fall into poverty temporarily”

3. I suggest changing the language of “significant fraction” – it makes it sound large, but I think the authors are saying the fraction is still generally small.

Section 1

4. On page 4, where authors say “no household member contributes to social security,” social security can mean different things in different countries. Would “contributory social protection programs” be a more accurate description of what the authors are referring to?

Discussion (conclusion)

5. The discussion section should be expanded.

a. Can the authors generalize about coverage and replacement rates across national income classifications (et, upper middle income, high income, etc.).

b. What can the authors say about budgets for these expansions, including where budgets came from (borrowing, which sectors, etc.), and the percentage of the new programs as compared to overall spending in an average year on social protection?

c. Can the authors suggest some mechanisms to expand coverage to informal workers who “who are not sufficiently poor to bene_t from social assistance but lack other automatic stabilizers”

6. PLOS authors have the option to publish the peer review history of their article (what does this mean?). If published, this will include your full peer review and any attached files.

Reviewer #2: **Yes: **Delia Boccia

Reviewer #3: No

---

## [Editor Report · Decision Letter 1]

12 Oct 2021

Social Protection and Informality in Latin America during the COVID-19 Pandemic

PONE-D-21-12981R1

Dear Dr. Messina,

We’re pleased to inform you that your manuscript has been judged scientifically suitable for publication and will be formally accepted for publication once it meets all outstanding technical requirements.

Kind regards,

Tom E. Wingfield

Academic Editor

PLOS ONE
---

## [Editor Report · Acceptance letter]

26 Oct 2021

PONE-D-21-12981R1 

Social Protection and Informality in Latin America during the COVID-19 Pandemic 

Dear Dr. Messina:

I'm pleased to inform you that your manuscript has been deemed suitable for publication in PLOS ONE. Congratulations! Your manuscript is now with our production department. 

Kind regards, 

on behalf of

Dr. Tom E. Wingfield 

Academic Editor

PLOS ONE